# Translating the Hypoxic Response—The Role of HIF Protein Translation in the Cellular Response to Low Oxygen

**DOI:** 10.3390/cells8020114

**Published:** 2019-02-01

**Authors:** Iglika G. Ivanova, Catherine V. Park, Niall S. Kenneth

**Affiliations:** Institute for Cell and Molecular Biosciences, Faculty of Medical Sciences, Newcastle University, Newcastle upon Tyne, NE2 4HH, UK; iglika.ivanova@newcastle.ac.uk (I.G.I.); catherine.park@newcastle.ac.uk (C.V.P.)

**Keywords:** Hypoxia, HIF, Translation, Protein Synthesis, mRNA, RNA binding Proteins, RBP, Micro RNAs, MiR

## Abstract

Hypoxia-Inducible Factors (HIFs) play essential roles in the physiological response to low oxygen in all multicellular organisms, while their deregulation is associated with human diseases. HIF levels and activity are primarily controlled by the availability of the oxygen-sensitive HIFα subunits, which is mediated by rapid alterations to the rates of HIFα protein production and degradation. While the pathways that control HIFα degradation are understood in great detail, much less is known about the targeted control of HIFα protein synthesis and what role this has in controlling HIF activity during the hypoxic response. This review will focus on the signalling pathways and RNA binding proteins that modulate HIFα mRNA half-life and/or translation rate, and their contribution to hypoxia-associated diseases.

## 1. Introduction

Respiration in human cells strictly depends on balancing oxygen levels to support aerobic respiration, without producing excessive reactive oxygen species that can result in damage to organelles [1,2]. An increase (hyperoxia) or decrease (hypoxia) in cellular oxygen levels, leads to the activation of cellular signalling pathways that allow cells and organisms to maintain oxygen homeostasis [1,2]. Oxygen levels are therefore tightly regulated in cells and tissues. However, hypoxia and the activation of hypoxia-induced cellular signalling pathways are a common denominator in the pathophysiology of human diseases [1,3]. As such, insight into how human cells detect and respond to low oxygen is crucial to understanding the role of the hypoxic response in disease. 

## 2. Hypoxia-Inducible Factors

Central to the hypoxic response are the Hypoxia-Inducible Factors (HIFs). HIFs are transcription factors essential for the adaptive response to low oxygen and the primary mediator of gene expression changes in hypoxic cells [1,2]. HIFs are activated in cells where oxygen levels fail to meet demand, at a point that is context dependent due to variations in oxygen requirements between cells and tissues within the human body (~100 mmHg in arterial blood to ~29 mmHg in muscle) [4]. This may be caused by chronic hypoxia due to oxygen deprivation, or cyclic hypoxia, wherein oxygen levels fluctuate significantly within the tissue microenvironment [5,6]. HIF was first recognised as a DNA-binding α/β-heterodimer that binds to an enhancer region of the human erythropoietin gene to promote its expression and stimulate red blood cell production [7]. HIF transcription factors are found in all human cell types and are recognised as key modulators of the transcriptional response to hypoxia [1,2]. HIF transcription factors play essential roles for the acute response to low oxygen in normal cells and tissues. However, deregulated HIF activity is frequently observed in malignant cells where it can induce changes in energy metabolism and protect cells from hypoxia-induced cell death [1,8,9,10]. To date, more than 100 direct HIF target genes have been identified, many of which have been shown to be involved in the control of the metabolic switch for optimal cellular adaptation to hypoxia, angiogenesis, energy metabolism, cell differentiation and apoptosis, all of which have important roles in normal cell function, but can contribute to disease pathogenesis [3,11,12]. 

The HIFα and HIFβ subunits are DNA-binding proteins that form transcriptionally active heterodimeric complexes to activate hypoxia responsive genes [3,9]. The human genome contains three HIFα subunits (HIF1α, HIF2α/EPAS, and HIF3α) and two HIFβ subunits (ARNT/HIF1β and ARNT2) [3,9]. HIF subunits share highly similar domain regions, characterised by the presence of bHLH (basic helix–loop–helix)–PAS (Per/ARNT/Sim) domains that mediate heterodimer formation and DNA binding [13]. In addition to these shared domains, HIFα subunits contain oxygen-dependent degradation (ODD) domains and transactivation domains (TAD) to promote the expression of target genes [14]. The two principle HIF complexes are comprised of HIF1β, and one of either HIF1α or HIF2α, which make up the transcription factors referred to as HIF1 and HIF2, respectively [1,9]. Despite their structural similarities and identical DNA recognition motifs, HIF1 and HIF2 are activated with different kinetics and bind to a distinct repertoire of cell-specific sites across the genome [15,16]. HIF3α cannot induce the expression of hypoxia-inducible target genes to the same extent as HIF1α and HIF2α as it lacks the C-terminal TAD [17]. HIF3α can, therefore, act as a suppressor of HIF-dependent gene expression by competing with HIF1α or HIF2α to bind HIF1β, or other binding partners for HREs in the promotors of target genes [1,17].

Targeted disruptions to either HIF1α or HIF2α in mice results in early embryonic lethality. However, the phenotypes are markedly different, with HIF1α^-/-^ mice dying by E11 with severe angiogenensis defects and HIF2α^-/-^ mice suffering from bradycardia, abnormal lung formation and blood vessel defects that kills the majority of embryos by E14 [18,19,20]. In comparison, HIF3α^-/-^ mice appear outwardly normal, and have relatively mild phenotype associated with heart development and lung remodelling [21]. The available data indicate that individual HIFα subunits play distinct and separate roles in physiology and disease [16]. This review will focus on the control of protein translation of the individual HIFα subunits, mainly concentrating on regulation of the HIF1α and HIF2α subunits, as a means of regulating HIF activity in response to hypoxic stress, and importantly, how HIF translation may be directed therapeutically to target-specific HIF complexes.

## 3. Control of HIF Activity by Ubiquitin-Mediated Proteolysis

In well-oxygenated cells, both HIF1α and HIF2α are continuously synthesised prior to being hydroxylated on conserved proline residues within their ODD domains by a family of prolyl-hydroxylase enzymes (PHD-1-4) [1,2,22]. The hydroxylated HIFα subunits are recognised by the von Hippel–Lindau (VHL) E3 ubiquitin ligase, which promotes the ubiquitination and subsequent degradation of HIF1α and HIF2α by the 26S proteasome1 [2]. When oxygen levels are reduced, the PHD enzymes are inhibited, resulting in HIF1α and HIF2α stabilisation and nuclear translocation [1,2]. When in the nucleus HIF1α and HIF2α can form heterodimers with HIFβ and activate the expression of hypoxia-responsive genes.

Much has been done to unravel the series of post-translational modifications required to control HIFα protein degradation (reviewed extensively [1,2]). However, considerably less is understood about the mechanisms that control HIFα subunit protein synthesis. Despite the identical modulation of HIF1α and HIF2α proteins by the PHD/VHL-dependent pathway, the available evidence suggests that HIF1 and HIF2 are disparately regulated within the cell [16]. Thus, alternative mechanisms must exist to independently control HIFα subunit levels in the cell. As regulation of protein abundance is ultimately controlled by the interplay between mRNA transcription, protein translation and protein degradation, intervention at each stage can define absolute protein levels. Control of HIF subunit mRNA translation is an important mechanism to control HIF activity, and represents a mechanism to independently control HIFα levels in hypoxic cells [23].

## 4. Control of Protein Synthesis in Hypoxia

Cellular control of mRNA translation may be global, in which the translation of the bulk of the mRNAs is regulated en masse, or mRNA specific, wherein the translation of defined subsets of mRNAs is modulated [24,25]. In cells and tissues exposed to hypoxia, disruptions to oxygen delivery results in decreased energy production and general suppression of energy expensive processes such as protein synthesis [2,9]. Under severe hypoxia, the ATP demand for protein synthesis drops to approximately 7% of normoxic cells, correlating with a dramatic reduction in protein translation rates [26]. Hypoxia-dependent repression of protein synthesis occurs mostly at the levels of translation initiation, generally considered to be the rate limiting step of protein translation [25]. Initiation occurs as the small ribosome subunit is recruited to the 5′ end of mRNA and scans towards the start codon [25]. Hypoxia prevents eukaryotic translation initiation via two distinct pathways propagated by the mammalian target of rapamycin (mTOR) [27,28,29] and the stress responsive protein kinase R (PKR)-like endoplasmic reticulum kinase (PERK) [30,31,32].

mTOR inhibition during hypoxia requires the action of the TSC1/TSC2 tumour suppressor complex and the hypoxia-inducible gene REDD1 [27,33]. Hypoxia triggers the release of the mTOR inhibitor TCS2 from 14-3-3 binding proteins in a REDD1-dependent manner to block mTOR activity, resulting in the hypophosphorylation of its effectors 4E-BP1, p70S6K and RPS6, ultimately leading to the inhibition of translation initiation [34,35,36]. This inhibition terminates the recruitment of the eukaryotic translation initiation factor 4F (eIF4F) complex to the 5′ cap structure of the mRNA transcript, culminating in translation block [34]. Interestingly, fibroblasts isolated from either TSC2 or REDD1-null mice exhibit increased proliferation rates and anchorage-independent growth under hypoxia as a result of aberrant mTOR activity, suggesting that loss of mTOR-dependent control of protein synthesis in hypoxia is tumour promoting [27,33].

In addition, severe hypoxia can lead to the phosphorylation and activation of PERK, one of a family of kinases that can phosphorylate and inhibit eukaryotic translation initiation factor 2 alpha (eIF2 α) [36]. Phosphorylated eIF2 α acts as a dominant inhibitor of eIF2B, which effectively prevents the recycling of eIF2 between successive rounds of protein synthesis, thus causing a general suppression of translational initiation and global protein synthesis [36]. Severe hypoxia results in the PERK-dependent phosphorylation of eIF2 α in several different mouse and human cell lines [30,37]. Genetic evidence from PERK-null mice demonstrate its protective role during hypoxic stress. MEFs isolated from PERK-null mice were more sensitive to hypoxic stress than matched wildtype cells, indicating the importance of PERK-dependent signalling during the hypoxic response [30,31,37].

The evidence suggests that the HIF-, mTOR- and PERK-dependent responses to hypoxia act in an integrated way, impacting each other and the hypoxia-dependent signalling pathways that affect gene expression, metabolism, cell survival, tumorigenesis and tumour growth [29,38].

## 5. Specific Control of HIF Protein Translation in Hypoxia

Repression of global protein synthesis, while necessary for the adaptive response to the changing energy conditions in hypoxia, is insufficient to trigger the adaptive response to restore oxygen homeostasis [39,40,41,42]. Overcoming translational repression during hypoxia is needed for de novo synthesis of proteins essential for the adaptive response to low oxygen. Translation of the HIF family of transcription factors and other proteins necessary for the hypoxic response is maintained to ensure cells have the correct repertoire of stress-responsive factors [23]. The specific control of mRNA-specific translation is dependent on signals encoded within *cis*-elements present in the mRNA’s 5′- or 3′-untranslated regions (UTRs).

The preferential translation of HIFα subunits during hypoxia was originally attributed to an internal ribosome entry site (IRES) encoded within the 5′ UTR of mRNAs [43,44]. IRES sequences allow specific mRNAs to be translated independently of the global rates of protein synthesis as it serves as a ribosome assembly site in the middle of the mRNA, which is independent from the eIF4F cap-binding complex [45]. However, subsequent studies have indicated that the HIF UTRs do not possess IRES activity [46,47], instead regulatory non-coding RNAs and/or RNA-binding proteins together regulate translation rates via interaction with the 5′ and 3′ UTRs of HIF subunit mRNAs [48,49,50,51].

### MicroRNAs

MicroRNAs (MiRs) are small noncoding RNAs that extensively regulate gene expression in animals, plants and protozoa [52,53]. Cellular stresses such as hypoxia induce the expression of a MiR signature that contributes to the changes in the proteome to restore oxygen homeostasis (Reviewed in [54,55,56]). Interestingly, the pattern of MiR activation under low oxygen is dependent on the type of hypoxic stress, with chronic hypoxia having a different signature to cyclic or intermittent hypoxia, adding an additional level of complexity to the system [57]. MiRs can directly post-transcriptionally regulate HIF transcription factors by binding to the mRNA 3’-untranslated regions reducing their half-life and/or inhibiting their translation [56,58,59]. Several MiRs have been described that bind directly to the 3′ UTR of the HIF1α transcript to suppress HIF1α levels, including MiR-155 [60], MiR-429 [61,62], MiR-519c [63], MiR-17-92 cluster [64,65,66], MiR-153 [67], MiR-199a [68], MiR-150 [69] and MiR-497 [70] (detailed in Table 1). MiR-155, MiR-429 and MiR-153 are induced in low oxygen as part of the hypoxia-induced MiR signature, perhaps representing an isoform-specific negative feedback loop for the resolution of HIF1α activity in cells exposed to prolonged hypoxia [60,61,62,67]. Although the other identified MiRs do not seem to be induced by hypoxia, they can directly alter HIF1α expression and reduce HIF-dependent angiogenesis and tumour survival in multiple cancer cell types including lung, breast and gastric carcinoma cell lines [63,64,65,66,67,68,69,70,71], representing an isoform-specific control of HIF isoforms in malignant cells.

A distinct subset of MiRs targeting HIF1β, HIF2α and HIF3α have been identified, mainly in tumour cell lines. However, their regulation appears to be hypoxia-independent [62,72,73,74] (Table 1). MiR-145, MiR-30a and MiR-30c can directly target the HIF2α 3′ UTR, suppressing HIF2α without altering the levels or activity of HIF1α, again demonstrating that differential MiR expression may represent a mechanism to control isoform-specific regulation of HIF transcription factors [72,73]. The difference in the numbers of MiRs identified targeting HIF1α mRNA, rather than HIF2α mRNA, more than likely represents the greater research focus on the activity of the HIF1α subunit, as the HIF2α mRNA’s 3′ UTR is approximately double the length of the HIF1α mRNA 3′ UTR (2064bp compared to 1197bp), suggesting additional HIF2α -specific MiRs are yet to be identified.

## 6. RNA-Binding Proteins and Their Role in HIF Subunit Translation

In addition to the encoding information that mediates the RNA/RNA interactions essential for the MiR-dependent control of HIF activity, the 5′ and 3′ UTRs of HIF subunits also form structures that can mediate interactions with RNA-binding proteins (RBPs) that profoundly affect their translation rates in human cells (Figure 1). Outlined below is a selection of the RNA binding proteins that can influence HIF subunit translation rates.

### 6.1. Y Box Binding Protein 1

Y box binding protein 1 (YB-1) is a highly conserved cold shock domain (CSD) family protein that can bind to both DNA and RNA [75,76]. YB-1 participates in a wide variety of DNA/RNA-dependent events including DNA repair, transcription, mRNA splicing and packaging, and regulation of mRNA stability and protein translation [75,76]. Unbiased mapping of the RNA species that interact with YB-1 in vivo reveal that the vast majority of YB-1/mRNA interactions occur either in the protein encoding regions or the 3′ UTRs of mRNAs and are thought to facilitate storage and repression of mRNA translation [77]. YB-1 can also induce the expression of certain stress-responsive factors by binding to the 5′ UTR of mRNAs to increase their translation rates [78,79]. One such mRNA that is positively regulated by YB-1 binding to its mRNA is HIF1α. YB-1 directly binds to a region within 5′ UTR of HIF1α to promote its translation [80]. The YB-1-HIF1α mRNA interaction is enhanced in hypoxic cells, creating a feed forward mechanism to maintain HIF1α translation in response to low oxygen [81]. HIF1α mRNA translation rates are dependent on the YB-1 interaction in hypoxic cells. Our lab has found that the YB-1/HIF1α mRNA interaction is negatively regulated by the activation of protein kinase, PERK, in moderate hypoxia, suggesting that activation of the UPR can have different effects depending on the severity of hypoxic stress [81]. Like the HIF proteins themselves, YB-1 expression has been correlated with the progression or severity of neoplastic diseases including lung, breast, gastric and colon cancers (reviewed in [82]). As high HIF levels and activity are characteristic of many of these tumour types, the oncogenic activity of YB-1 may be due to its role in regulating the HIF pathway.

### 6.2. Hu Antigen R

Hu Antigen R (HuR) is a ubiquitously expressed RNA-binding protein involved in the regulation of translation by directly binding to the conserved RNA sequence motifs of a variety of stress-responsive proteins [83]. HuR binds to a large number of mRNAs bearing AU- and U-rich UTR sequences and transcriptome wide analysis reveals ~26,000 endogenous HuR binding sites, binding up to 4874 mRNA sequences, representing about half the transcripts in a HeLa cell [51,83]. HuR knockdown leads to a highly significant destabilisation of transcripts with HuR binding sites, confirming its role in regulating mRNA stability [51]. HIF1α mRNA, but not any other HIF subunit, has been identified as a HuR target in both candidate-based [84,85] and unbiased screens [51]. HuR can bind to both the 5′ and 3′ UTRs of the HIF1α mRNA in human cells [84,85]. HuR binding to the 5′ UTR of HIF1α is surprising as this region of HIF1α mRNA is extremely GC-rich. However, experimental evidence suggests the overexpression of HuR dramatically increases the translation of a reporter transcript fused to the HIF1α 5′ UTR sequence. Interestingly, no observable difference is seen in translation rates using a 3′ UTR-containing reporter construct [85]. The link between HuR promoting HIF1α mRNA translation is supported in subsequent studies examining the pro-oncogenic properties of HuR in meningioma, in which HuR depletion suppresses the HIF1 signalling pathway [86].

### 6.3. Polypyrimidine Tract Binding Protein

Like YB-1 and HuR, the polypyrimidine tract binding protein (PTB) can regulate multiple steps in the mRNA lifecycle from splicing, localisation and storage to translation [87,88]. PTB is ubiquitously expressed in mammalian cells and binds with high affinity for tracts of polypyrimidine (CU) – motifs in RNA [87,89]. PTB has been characterised as an enhancer of HIF1α translation through binding to the HIF1α UTR in cultured human kidney cells in a manner similar to YB-1 [43]. Further studies found that instead of exclusively binding to the 5′ UTR of HIF1α, PTB could bind to both the 5′ and 3′ UTR to promote HIF1α translation [85]. PTB homodimers contain four RNA-binding domains (RBDs); the role of PTB may only be clear in the context of the whole mRNA sequence rather than in reporter assays. Structural studies have shown that the RBDs of PTB each bind with a subtly different binding specificity, allowing a single PTB molecule to create RNA loops in a target mRNA, facilitating splicing, or modulating the mRNA structure [89]. Due to the presence of PTB binding sites in both the 5′ and 3′ UTR of the HIF1α mRNA, it is possible that PTB, along with other HIF1α mRNA RBPs, can create mRNA loops to facilitate ribosome recycling [90]. Although many of these studies have examined the effects of RNA binding proteins in isolation, the available evidence suggests that a single HIF1α mRNA can bind to multiple RBPs. Indeed, there appears to be co-operation between PTB and HuR in promoting HIF1α translation in reporter assays [85].

### 6.4. Tristetraprolin

Additional control of the HIF1α 3′ UTR is provided by the Tristetraprolin (TTP) family of RNA binding proteins [91,92]. The human TTP family consists of three members, TTP (TIS11/ZFP36), TIS11b (ZFP36L1/BRF1) and TIS11d (ZFP36L2/BRF2), which own the same characteristic CCCH tandem zinc-fingers and share similar mRNA-destabilising activity in vitro [93,94]. The overexpression of each of the three family members can suppress the expression of a HIF1α 3′ UTR reporter construct through the control of mRNA stability [92]. TTP family members bind to AU-rich elements within the UTRs of a variety of genes and endogenous TTP physically interacts with HIF1α mRNA as measured by RNA IP [92]. In the case of the HIF1α transcript, TTP can bind to a well-defined region in the 3′ UTR of HIF1α and induce its degradation [91,92]. siRNA knock down experiments show that TTP can specifically destabilise HIF1α mRNA in cells exposed to prolonged hypoxia, but not normoxic cells, perhaps indicating that TTP plays a role in a hypoxia-specific HIF1 negative feedback loop [91,92]. The mRNAs targeted by TTP encode protein products such as HIF1α, which are critical for the progression of several malignancies [95,96,97,98]. The loss of TTP has been reported in several human cancers; this correlates with the elevation of HIF1α and poor prognosis [95,96,97,98].

### 6.5. RNA Binding Motif Protein 38

RNA Binding Motif Protein 38 (RBM38) is an additional RNA-binding protein that associates with the AU-rich elements in the 3′ UTR of the HIF1α mRNA [99]. Like PTB, RBM38 can also bind to the GC-rich 5′ UTR of HIF1α mRNA as demonstrated by EMSA and RNA IP [99]. RNA IP and EMSA show that the binding of RBM38 to the 5′ UTR and 3′ UTR of HIF1α mRNA prevents the association of eIF4E with the 5′ cap of the HIF1α mRNA [99]. Moreover, RBM38 provides a link between the p53 tumour suppressor and the hypoxia signalling pathway. RBM38 expression is under direct control of p53. Therefore, the loss of p53 diminishes the inhibitory effect of RBM38 on HIF1 activity to promote the malignant phenotype [99,100]. This is another example of how the loss of p53 can promote tumorigenesis by activating the HIF pathway to promote angiogenesis and changing energy metabolism.

### 6.6. Cytoplasmic Polyadenylation Element Binding Proteins 1 and 2

Cytoplasmic Polyadenylation Element Binding (CPEB) proteins bind to cytoplasmic polyadenylation elements (CPEs) in the 3′ UTRs of specific mRNAs to regulate poly (A) tail growth or removal, which promotes or represses translation. CPEB1, and to a lesser extent CPEB2, bind the CPE-containing fragment at the very 3′ end of the HIF1α mRNA [101]. Overexpressing CPEB1 and 2 decreases HIF1α protein levels and alters the expression of HIF-target gene expression in cultured cells, indicating that the CPEB1 can alter HIF1α translation rates in cells [101]. Reduced levels of CPEB1 are associated with several types of cancer, cell invasion and angiogenesis, processes which correlate with high HIF activity [102,103].

### 6.7. Cold-Inducible RNA Binding Protein

A recent study has described a role for the cold-inducible RNA-binding protein (CIRBP) as a novel modulator of HIF1α translation in human bladder cancers. CIRBP has been reported to be pro-tumorigenic in a number of human cancers [104]. CIRBP binds specifically to the 3′ UTR of the HIF1α transcript to increase its stability and elevate HIF1α protein synthesis [104]. The depletion of CIRBP using RNAi suppressed HIF1α levels to reduce the proliferation and migration of bladder cancer cell lines. This again provides further evidence of how inappropriate expression-specific RBPs can alter HIF activity in disease.

### 6.8. Iron Regulatory Proteins

Iron is an essential element in all living organisms and is required as a cofactor for oxygen-binding proteins. Iron metabolism, oxygen homeostasis and erythropoiesis are consequently strongly interconnected, with iron acting as an essential cofactor of many oxygen-binding proteins [105]. Iron regulatory proteins (IRPs) control iron metabolism by binding to specific non-coding sequences within an mRNA, known as iron-responsive elements (IRE). IREs are 30 nucleotide long RNA motifs that form special stem-loop structures that reside in the 3′ UTR and 5′ UTRs of an mRNA. Analysis of the HIF2α mRNA revealed a conserved, functional iron-responsive element (IRE) in its 5′ UTR [106]. Low concentrations of iron increase the binding of IRP1 to the 5′ UTR of the HIF2a mRNA, repressing HIF2α translation rates and protein levels [106,107]. The disruption of mouse IRP1 leads to profound HIF2α-dependent abnormalities in erythropoiesis and systemic iron metabolism [108].

## 7. Direct Regulation of Protein Translation by HIF2α

Many studies have demonstrated that translational control of the HIF subunits controls the length and intensity of the HIF-dependent hypoxic response. However, a recent study has suggested that HIF2α itself can act as a direct regulator of the translational rates of a subset of hypoxia-induced target genes [33]. Acting as part of a tripartite structure composed of the RNA binding protein, RBM4, and the translational regulator, eIF4E2, HIF2α captures the 5′cap and targets specific mRNAs, such as the EGFR mRNA to polysomes [33]. This represents a previously unanticipated role of HIF transcription factors playing a direct role in shaping the cellular proteome in response to hypoxic stress.

## 8. Control of Messenger RNA Subcellular Localisation

The localisation of mRNA is coupled to translational regulation and can provide an important means of controlling the expression of the cellular proteome. The localisation of mRNAs to discrete subcellular locations in the cytoplasm can have profound effects on the translation rates of specific mRNAs [109,110]. Control of mRNA subcellular localisation, once considered a specialised mechanism restricted to a very small fraction of transcripts controlling cell polarity or migration, is emerging as a conserved process regulating mRNA translation in a variety of cellular events. Studies performed in hypoxic yeast cells indicate a rapid subcellular redistribution of RNA-associated proteins involving mRNA splicing, cleavage and processing [111]. The unbiased measurement of mRNA localisation in hypoxic human fibrosarcoma cells reveals a shift in HIF1α transcripts to the ER-associated ribosomes upon the induction of hypoxic stress compared to normoxic controls [112]. The ER is a privileged site of protein synthesis in stressed cells, as such global rearrangement of the mRNAs represents a mechanism to switch the translational profile of stressed cells. Increased levels of ER-associated transcripts were observed in hypoxic cells [112], indicating the presence of conserved stress-responsive sequences within the UTRs of mRNAs that allow the co-ordinated expression of the hypoxia responsive proteome in low oxygen [112]. The mechanisms that direct mRNA partitioning by *cis*-element/*trans*-factor interaction during hypoxia are unclear. However, as YB-1, HuR and PTB can rapidly relocalise in conditions of cell stress [78,113], it is tempting to speculate that the relocalisation is, in part, due to the UTR RBPs discussed earlier in this review.

## 9. HIF Translation—A Potential Therapeutic Target?

As hypoxia and key hypoxia effectors such as HIF contribute to the pathologies of several human diseases, there is a great deal of interest in developing novel therapeutics to modulate HIF activity [1]. Small molecule inhibitors of the PHD and VHL enzymes that can hyperactivate HIF signalling have been developed to offer potential therapeutics for conditions where lack of HIF activity is pathogenic [2,114,115]. Currently, several PHD inhibitors are in clinical trials to enhance HIF activity in anaemic patients to boost red blood cell production [116]. The development of small molecules that inhibit aberrant HIF activity, particularly in the context of cancer therapeutics, is complicated by the conflicting roles of HIF1 and HIF2 in certain tumour types [117]. Optimal HIF inhibitors would ideally allow differentiation between the HIF1α and HIF2α subunits. However, existing molecule inhibitors targeting HIF dimerisation, nucleic acid binding and transcriptional activity alter the activity of both HIF1 and HIF2 [118].

As discussed in this review, HIF1α and HIF2α are regulated by a distinct set of translational activators, offering an opportunity to differentiate between HIF1 and HIF2 therapeutically. Indeed, a number of structurally and functionally distinct drugs capable of modulating HIF translation have been identified that appear to have preference for specific subunit complexes. In our lab, we have described how the sarco/endoplasmic reticulum Ca^2+^ ATPase (SERCA) inhibitor thapsigargin disrupts the association of the RBP, YB-1, with HIF1α mRNA specifically repressing HIF1α levels and activity without altering HIF2α [81]. Conversely, arsenite enhances HIF1 activity by inhibiting the binding of CEPB2 to the 3′ UTR of HIF1α mRNA, enhancing the production of the protein [119]. In addition to its role in inhibiting the PHD enzymes, CoCl_2_ also enhances HIF1α protein production through stimulating the association of HuR and PTB to the UTRs [85].

An unbiased screen using a HIF reporter cell line identified an array of cardiac glycosides that can inhibit HIF1α translation independently of oxygen level in a UTR-dependent manner [120]. In addition, the DNA alkylating agent mitomycin C, the topoisomerase inhibitor camptothecin, and the anti-clotting agent YC-1 also decrease HIF1a protein levels through the 5′ UTR of its mRNA without affecting mRNA stability [121].

The HIF2α mRNA/IRP1 interaction can also be exploited pharmacologically to specifically modulate HIF2 activity, without altering the levels and activity of HIF1α. An unbiased screen to identify novel HIF therapeutics identified selective inhibitors of HIF2α translation [106]. Hypoxia derepresses HIF2α translation by disrupting the IRP1/HIF2α mRNA interaction [122]. The small molecule identified in this study promotes and restores the IRP/HIF2α mRNA interaction to specifically repress HIF2 without altering HIF1 activity [107,122].

## 10. Perspective

Since its discovery over 25 years ago, great advances have been made in our understanding of the role of the HIF family of transcription factors in normal physiological responses and disease. The series of post-translational modifications that control the oxygen-dependent degradation of the HIFα subunits is well-defined and understood. However, it is becoming clear that targeted control of protein synthesis is critical for controlling HIF levels and activity in response to low oxygen. As we gain deeper knowledge of the how the RBPs and non-coding RNAs control HIF mRNAs, their potential usefulness as therapeutic targets is becoming apparent. As the factors controlling HIF1α and HIF2α mRNA are distinct, this offers the opportunity to target these complexes separately. Advances have been made in identifying several modulators of HIF translation rates. However, much of this work has relied on reporter assays rather than the direct measurement of translation rates. Careful, systematic, analysis of the role of RNA binding proteins and MiRs that direct HIF subunit translation is needed if these pathways are to be clinically exploited.

## Figures and Tables

**Figure 1 cells-08-00114-f001:**
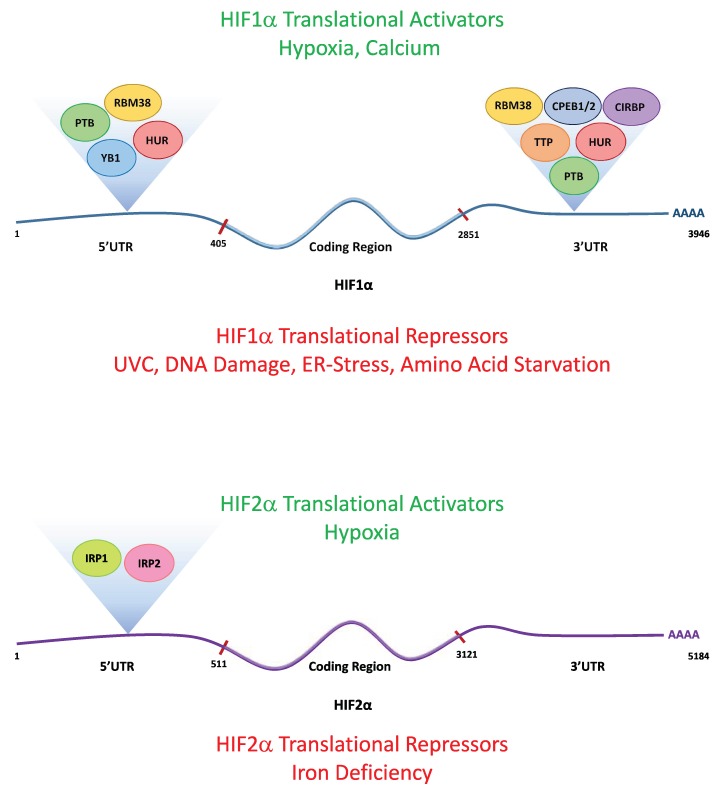
Schematic of the HIFα mRNAs and their RNA-binding proteins (RBPs). Schematic showing the known HIF1α and HIF2α RBPs, the region of the mRNA they are known to bind to and the known physiological modulators of HIF1α and HIF2α translation rates. The positions of the start of the 5’ UTR, coding region and the 3’ UTR are taken from the nucleotide sequences NM_001530 (HIF1A) and NM_001430.5 (HIF2A).

**Table 1 cells-08-00114-t001:** MicroRNAs (MiRs) that directly degrade Hypoxia-Inducible Factor (HIF) subunit mRNA.

Target mRNA	MiR	Response to Hypoxia	MiR Binding Site(s)	Reference
HIF1α	MiR-17-5p	Repressed	3520/3738	57, 58, 59
HIF1α	MiR-18a	Repressed	3042	57, 58, 59
HIF1α	MiR-20a	Repressed	3027/3190	57, 58, 59
HIF1α	MiR-20b	Repressed	3737	57, 58, 59
HIF1α	MiR-153	Induced	3446	60
HIF1α	MiR-155	Induced	3799	53
HIF1α	MiR-199a-5p	Not Tested	2810	61
HIF1α	MiR-210	Induced	2884	50, 63
HIF1α	MiR-429	Induced	3218	54, 55
HIF2α	MiR-30a-3p	Not Tested	4993	67
HIF2α	MiR-30c-2-3p	Not Tested	4006	67
HIF2α	MiR-145	Not tested	3919	66
HIF3α	MiR-429	Induced	4301	55
HIF1β	MiR-107	Induced	3361	50, 68

Indicated in the table is the MiR, the specific HIF substrate targeted, the position within the mRNA that the MiR binds and if the MiR expression level itself is regulated by low oxygen. All presented numbers are based on the nucleotide sequence of NM_001530 (HIF1A), NM_001430.5 (HIF2A), NM_152794.3 (HIF3A) and NM_001668.4 (HIF1B) and represent the nucleotide position in the mRNA that binds to the 5’ nucleotide of the MiR.

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
