# Peer review of "Translating the Hypoxic Response—The Role of HIF Protein Translation in the Cellular Response to Low Oxygen"

_cells, 2019, doi:10.3390/cells8020114_

Round 1
Reviewer 1 Report
This is an excellent review on an area relatively ignored in the filed but of great importance. It will be a point of reference for the future.
My only suggestion is a diagram including how different processes alter HIF-1alpha translation, to help with the text.
Author Response
Reviewer 1
This is an excellent review on an area relatively ignored in the filed but of great importance. It will be a point of reference for the future.
My only suggestion is a diagram including how different processes alter HIF-1alpha translation, to help with the text.
This is an excellent suggestion and have included in Fig 1 the known physiological modulators HIF subunit mRNA translation. (Modified Figure 1)
Reviewer 2 Report
I am convinced that the presented manuscript provides interesting insight into HIFs regulation during the hypoxia on protein levels. However I have to rise couple of points:
Major:
1. Authors should clearly distinguish and discuss the difference between hypoxia in normal endothelial cells and cancer cells and its impact on HIFs
2. Following point one - cancer cells are usually subjected to cyclic hypoxia having different effects on HIFs levels - please see Kochan et all 2019 - cyclic hypoxia effects on HIFs should be mentioned
3. What about KLF2 impact on HIFs protein stability
4. What about HIF3?
5. The miRNA sections - it's very limited and that's actually fair - but it would be good to provide some actual reviews papers like https://doi.org/10.1007/s10456-017-9545-x and https://doi.org/10.1007/s10456-018-9600-2
6. Authors should discuss the differences between the in vitro and in vivo studies on their subjects
7. Authors could provide critical discussion of unresolved problems.
Author Response
Reviewer 2
I am convinced that the presented manuscript provides interesting insight into HIFs regulation during the hypoxia on protein levels. However, I have to rise couple of points:
Major:
1. Authors should clearly distinguish and discuss the difference between hypoxia in normal endothelial cells and cancer cells and its impact on HIFs
We think this is an important point raised by the reviewer and have included extra information in our introduction to highlight the different roles of HIF in normal, compared to tumour cells. (Lines 52-60)
2. Following point one - cancer cells are usually subjected to cyclic hypoxia having different effects on HIFs levels - please see Kochan et all 2019 - cyclic hypoxia effects on HIFs should be mentioned
This is an interesting point made by the reviewer, we have now included additional discussion in the introduction and the MiR section addressing intermittent/ or cyclic hypoxia. (Lines 46-48 and 179-184)
3. What about KLF2 impact on HIFs protein stability
The role for KLF2 in regulating HIF1a protein stability is through its ability to control HIF1a stabilisation, albeit independently from the well-described VHL-dependent pathway. Although KLF2-dependent suppression of HIF1a levels is clearly important, I feel that it is not within the scope of this article.
4. What about HIF3?
We have now expanded on the description of HIF3a and have contrasted its function with that of HIF2a and HIF3a. (Lines 74-90)
5. The miRNA sections - it's very limited and that's actually fair - but it would be good to provide some actual reviews papers like https://doi.org/10.1007/s10456-017-9545-x and https://doi.org/10.1007/s10456-018-9600-2
This is an excellent suggestion by the reviewer, and we have added these additional references to the MiR section (Lines 179-184)
6. Authors should discuss the differences between the in vitro and in vivo studies on their subjects
7. Authors could provide critical discussion of unresolved problems.
As a response to points 6 and 7 jointly, the majority of the studies described in this article use in vitro reporter assays to assess the roles of each of the RBPs on HIF subunit translation. Critical to the advancement of this field further work needs to be done to examine the role for controlled HIF subunit translation by direct measurement of translation. Additional discussion explaining this has been added to the perspectives section. (Lines 421-425)
Reviewer 3 Report
In this manuscript Ivanova and colleagues describes the effects of RNA binding proteins implicated in the hypoxic response and their role in regulating HIF mRNA. The review is generally well written and easy to follow. I have some suggestions that need to be taken into account:
1) In the description of the hypoxia inducible factors (section 2) the authors focus on HIF-1α and HIF-2α. HIF-3α knockout mice are missing in this section and they are probably worth mentioning to point out in more detail the differences between HIF1/2α and HIF3α, because there are viable and show a relative mild phenotype
2) Figure 1: the authors should not use the same color for different RBPs in order to make them more visually pleasing for the reader.
3) Page 3, line 102: after “in translation block” a ref. is needed.
4) Page 6, line 204: “in human cells” is written twice.
5) Page 7, line 267: after “HIF1α protein synthesis” a ref. is needed.
Author Response
Reviewer 3
In this manuscript Ivanova and colleagues describes the effects of RNA binding proteins implicated in the hypoxic response and their role in regulating HIF mRNA. The review is generally well written and easy to follow. I have some suggestions that need to be taken into account:
1) In the description of the hypoxia inducible factors (section 2) the authors focus on HIF-1α and HIF-2α. HIF-3α knockout mice are missing in this section and they are probably worth mentioning to point out in more detail the differences between HIF1/2α and HIF3α, because there are viable and show a relative mild phenotype
As discussed in our response to Reviewer 2, I think this is an excellent suggestion and have included the details of the HIF3a knockout mouse and compared and contrasted its phenotype with the HIF1a and HIF2a mice. (Lines 75-90)
2) Figure 1: the authors should not use the same color for different RBPs in order to make them more visually pleasing for the reader.
All Different RBPs are now a separate colour
3) Page 3, line 102: after “in translation block” a ref. is needed.
This has now been addressed (Line 137)
4) Page 6, line 204: “in human cells” is written twice.
This has now been addressed
5) Page 7, line 267: after “HIF1α protein synthesis” a ref. is needed.
This has now been addressed (Line 323).